# Novel Low-Speed Measuring Method Based on Sine and Square Wave Signals

**Qiankun Song, Jigou Liu and Yang Liu *** 

ChenYang Technologies GmbH & Co. KG, Markt Schwabener Str. 8, 85464 Finsing, Germany
* Correspondence: yang.liu@chenyang.de; Tel.: +49-8121-2574102

**Abstract:** This paper presents a novel low-speed measuring method using analog sine and square waves of Hall effect speed sensors coupled with correlative digital signal processing algorithms packaged on a signal processing unit. The frequency of the initial signal is estimated by a square wave period measuring method (SWPM). On the basis of the initially measured frequency, a recursive self-correction (RSC) algorithm is used to perform the low-frequency measurement using the discrete sinusoid wave. The low-speed signal frequency can be derived continuously from the phase difference of the discrete sine wave, where the RSC algorithm is used to achieve high measuring accuracy. Compared to the method using only the SWPM algorithm, this novel low-speed measuring method enables faster measuring speed to achieve sufficient real-time performance. Simulation analyses and experiments verified the effectiveness of the proposed low-speed measuring method.

**Keywords:** low-speed measurements; Hall effect speed sensor; sine and square wave signals; analog-to-digital converters (ADCs); square wave period measuring method (SWPM); discrete Fourier series (DFS) algorithm; recursive self-correction (RSC) algorithm



## 1. Introduction

Low-speed rotating machines are essential in many production branches all around the world. These machines tend to be large and critical components in production lines. For example, they are in use at hydropower plants. Their moving elements such as bearings, gears, rotors, and gears of shafts, which are subjected to degradation over time, require sustained monitoring [1]. Thus, the rotational speed is an important control parameter which should be precisely controlled and monitored during the operation of low-speed rotating machines such as wind turbines. Most modern wind turbines transform the shaft's slow rotation into the fast rotation of a generator through a gearbox [2]. As an alternative, direct-drive machines, in which the turbine's shaft is directly coupled to the generator rotor, have become popular for offshore wind as they eliminate the necessity of a gearbox. However, direct-drive generators are much larger in size compared to their geared counterparts because of the high torque requirement at low speed, approximately 10–25 rpm [3]. When wind speeds are too high for safe operation, the turbine's rotor can be slowed down to prevent damage to the machine. Speed measurement of the turbine's low-speed shaft makes it possible to monitor the turbine system and ensure safe operation.

The measurement of the machinery's rotational speed is typically performed on the basis of mechanical adherence. Encoders are one example [2]. The encoders are electromechanical devices which give information about the angular position and the number of turns [4]. Although encoders are widely used in the field of speed measurement, high-resolution encoders are usually expensive and not suitable for cost-effective measuring systems. Laser Doppler velocimetry (LDV) is an alternative which relies on the Doppler effect of a laser beam to measure the vibration or velocity of a target. LDV achieves noncontact measurements with a very-high-frequency response [5]. The signal processing method for LDV systems usually uses the fast Fourier transform (FFT) to obtain the Doppler

frequency, which is a key factor in the calculation of an object's velocity. The Doppler effect is also utilized in "self-mixing" interferometry, which is also a popular research topic today. Compared to LDVs, "self-mixing" interferometers are more compact, less expensive, and suitable for low-speed measurement. However, the technology still faces many difficulties and, thus, requires plenty of research work before any practical application [6,7]. Therefore, using the Doppler effect is currently still complex and expensive; thus, it is not suitable for cost-effective measuring systems. Compared to encoders and LDVs, gear tooth speed sensors such as Hall effect sensors or optical reflective gear tooth sensors have a simpler structure and lower cost. Hence, Hall effect speed sensors are widely used in industry to provide large speed ranges and relative fast response times at low costs.

These measuring devices are divided into digital or analog according to output types. On digital outputs, the output can be binary coded, gray coded, or pulsed [4]. In practice, a digital tachometer is commonly used to measure the revolutions per minute (RPMs) [8]. Its measuring methods are usually based on timers/counters or analog-to-digital converters (ADCs). Two well-known speed estimation methods are considered: square wave frequency measuring (SWFM) and square wave period measuring (SWPM) methods. The SWFM method estimates the speed by counting the pulses of the pulse train during a fixed time interval. The SWPM method is based on counting the number of high-frequency clock pulses within the signal period. Considering pulse quantization errors, the SWFM and SWPM methods are best employed at high and low speeds, respectively [9,10]. Although the SWPM method provides high accuracy at low speeds, the response time of the measurement is limited by the target object's speed or the signal period. It takes a signal cycle to get a measuring result, which represents a large time delay at ultralow speeds. Therefore, the SWPM method cannot achieve sufficient real-time performance for low-speed measurements. To solve this issue, the ADC method for sinusoid signals and correlation digital processing algorithms such as recursive self-correction (RSC) and discrete Fourier series (DFS) are required.

The paper presents a novel method for low-speed measurement with high accuracy and high real-time performance, which is based on sine and square waves of the same frequency generated simultaneously by a speed sensor. The method can be used to build cost-efficient systems for low-speed measurements with high performance. The characteristics of the new method compared to the other speed measuring methods described above are shown in Table 1.

**Table 1.** Characteristics of different speed measuring methods.

| Feature | Encoder | LDV | Gear Tooth Speed Sensor (SWFM/SWPM) | Novel Measuring Method |
|---|---|---|---|---|
| Principle | Optical/magnetic | Laser Doppler effect | Optical/magnetic | Optical/magnetic |
| Structure | Complex | Complex | Simple | Simple |
| Accuracy | Very good | Very good | Good | Good |
| Robustness | good | Very good | Good | Good |
| Resolution | High | High | Low | High |
| Algorithm | SWPM/SWFM | FFT, etc. | SWPM/SWFM | RSC-DFS + SWPM |
| Cost | High | High | Low | Low |

The article is organized as follows: Section 2 describes low-speed measuring methods. These include the SWPM method, frequency measuring method based on phase difference, and the DFS and RSC algorithms, which are described in detail. In Section 3, the feasibility of the presented RSC-DFS method for sine signals is verified using MATLAB simulations. Subsequently, Section 4 describes a signal processing unit using the presented low-speed measuring method, where the measuring method is evaluated with a signal generator, as well as a Hall effect gear tooth sensor in a speed range of 2–100 RPM. Lastly, Section 5 provides some examples of potential low-speed measuring applications.

## 2. Low-Speed Measuring Method

### 2.1. Measuring Principle

In contrast to traditional rotational speed measurements, which are commonly based on square wave pulses, the new low-speed measuring method mentioned in this paper deals with original sinusoidal signals. This low-speed measuring principle is presented in Figure 1.

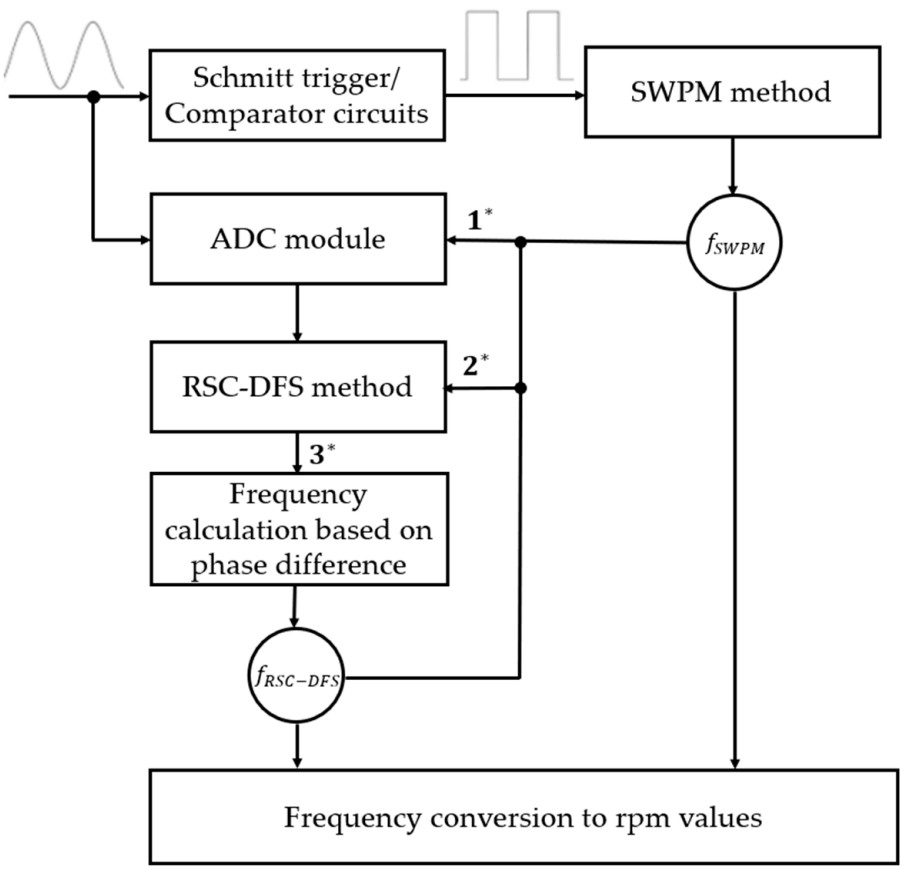

**Figure 1.** Schematic diagram of the low-speed measuring principle. (1*: Set the ADC sampling frequency on the basis of the frequency value. 2*: Determine the significant parameter $N_{DFS}$ of the DFS algorithm. 3*: Phase values at two adjacent sampling points, i.e., $\Phi_i$ and $\Phi_{i+1}$ *with* $i \geq N_{DFS}$.)

By means of Schmitt triggers or comparator circuits, square signals can be generated from sine signals. In addition to hardware circuits, Schmitt triggers in software can also implement the conversion to square signals. The initial frequency of the signal can be estimated with the square wave periode measurement method (SWPM). Then, the sampling frequency of the ADC is set properly according to the signal's initial frequency, i.e., to initialize the ADC. The initial frequency value from the SWPM method is also used to determine the number of sampling points per signal period $N_{DFS}$, which is the most critical parameter for the DFS algorithm (see Section 2.4). After sampling, the original sine signal is converted into the discrete sine signal by the ADC module. Subsequently, the signal's phase value corresponding to each sample point after $N_{DFS}$ samples is calculated from the discrete signal using the RSC-DFS method. Finally, the frequency is calculated by the phase difference of two adjacent sampling points and the sampling period (see Section 2.3).

As shown in Figure 1, on the one hand, the calculated frequencies from the SWPM method and RSC-DFS method are fed back to the ADC module to adjust the sampling frequency. On the other hand, they are fed back into the RSC-DFS method to update the $N_{DFS}$. With this feedback mechanism, dynamic and cyclic frequency measurements are realized. Finally, the frequencies need to be converted to rotational speeds on the basis

of their relationship with each other. The subsections below describe in detail the SWPM method, the DFS algorithm, and the RSC algorithm.

### 2.2. Square Wave Periode Measurement (SWPM) Method

The timer is a special function module in the microcontroller (MCU). In control systems, a real-time clock is often needed to time or to control delays, such as timed interrupts, timed detections, and timed scans. In addition, counters are often required to count external events. Timers play a very important role in square pulse frequency measurements. Usually there are two methods for square wave measurement: square wave period measurement and square wave frequency measurement. For low-speed measurements, square wave period measurements achieve higher measurement accuracy.

The frequency measurement of a square wave is achieved indirectly via its period measurement (see Figure 2). This method can be implemented using the timer's input capture mode. The rising or falling edge of the square wave signal triggers the timer to start counting. This makes it possible to calculate the period of the square wave using Equation (1).

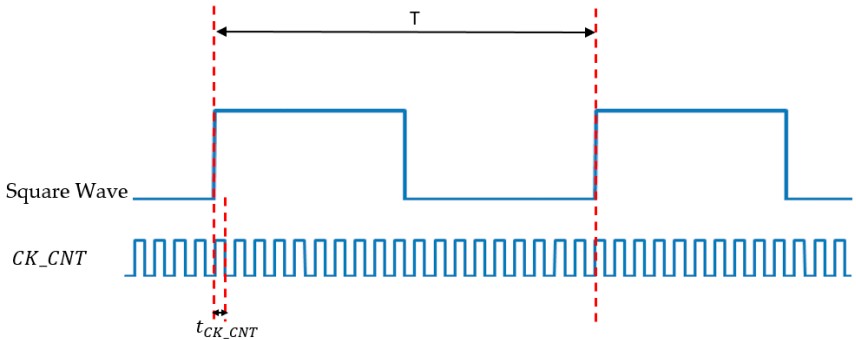

**Figure 2.** Schematic diagram of square wave period measurement method.

The number of cycles $CK\_CNT$ in period $T$ is $N_{CNT}$, from which the frequency $f$ can be derived [11]:

$$T = t_{CK\_CNT} \times N_{CNT},\tag{1}$$

$$f = \frac{1}{T} = \frac{1}{t_{CK\_CNT} \times N_{CNT}} = \frac{f_{CK\_CNT}}{N_{CNT}}.\tag{2}$$

The theoretical deviation $\varepsilon$ of the measuring result depends on the counter's clock frequency and the measured frequency, i.e.,

$$\varepsilon = \frac{t_{CK\_CNT}}{T} \times 100\% = \frac{f}{f_{CK\_CNT}} \times 100\%.\tag{3}$$

The theoretical deviation of the SWPM method is proportional to the measured signal frequency (or the rotational speed). Therefore, the SWPM method is suitable for low-speed measurements thanks to high measuring accuracy. Taking a Hall effect gear tooth as an example, according to Equation (21) in Section 4.2, if the number of gear teeth $N_r$ is set to 12 and the counter's clock frequency is set to 80 MHz, the theoretical measuring deviation is less than $2.5 \times 10^{-5}\%$ in the speed range of 2–100 rpm.

Despite the high measuring accuracy, the measuring principle is limited due to the long counting time $T$ of one signal period and does not meet the requirements for high real-time performance. The sinusoidal digital signal processing algorithms are consequently needed to improve the real-time performance for practical applications.

### 2.3. Frequency and Speed Measurement Based on Phase Difference Measurement

The quasi-sine speed signal can be described mathematically as a sine wave curve. Its simplest form as a function of time (*t*) is as follows [12]:

$$x(t) = A \sin(2\pi t f + \Phi) \quad with \ t \geq 0, \tag{4}$$

where *A* represents the amplitude, *f* is the frequency, and $\Phi$ is the phase. For sine signals, the total phase angle of a period is 360° (in radian $0 \leq \Phi < 2\pi$) and corresponds to the period duration *T*. The phase difference corresponding to two points within a period at an interval $\Delta t$ can be derived as follows:

$$\Delta\Phi = 2\pi f \Delta t \qquad with \ 0 \leq \Delta t < T. \tag{5}$$

From the above equation, the following frequency calculation formula can be derived:

$$f = \frac{\Delta\Phi}{2\pi\Delta t} \qquad with \ 0 \leq \Delta t < T. \tag{6}$$

The rotational speed can be obtained by conversion with respect to frequency. For digital signal processing, analog sine signals need to be converted into discrete digital signals by ADC sampling. It is assumed that $\Delta t$ is the known ADC sample period $t_s$. Therefore, the phase difference between two adjacent sampling points needs to be calculated for the frequency calculation. The discrete signal's phase can be determined by the DFS algorithm described next.

### 2.4. Discrete Fourier Series (DFS) Algorithm

In measurement technology, the discrete Fourier series (DFS) is often considered a special variant of the discrete Fourier transform (DFT) for periodic signals, which is widely used in information technology, as in the evaluation of electrical and non-electrical quantities. An example of application is the modeling of sinusodial signals for the measurement of the transfer function and the impedance of electrical systems. Signal modeling with DFS can filter the random noise [13].

A sinusoidal signal $x(k)$ sampled with a sampling period $t_s$ can be represented by the DFS using a fundamental waveform. The discrete signal $x(k)$, $k = 0, 1, \ldots, N_0 - 1$, within the period $T_0$ is described by the following equation [14]:

$$(k) = x(t_k) = \frac{a_0}{2} + c_1 \sin(2\pi f_0 t_k + \Phi_1). \tag{7}$$

By substituting $f_0 = \frac{1}{T_0} = \frac{1}{N_0 t_s}$ and $t_k = kt_s$ into the above equation, the expression of DFS for discrete signals is derived as follows [15]:

$$x(k) = \frac{a_0}{2} + c_1 \sin\left(2\pi\frac{k}{N_0} + \Phi_1\right), \tag{8}$$

with

$$c_1 = \sqrt{a_1{}^2 + b_1{}^2} \qquad \Phi_1 = \tan^{-1}\frac{a_1}{b_1}, \tag{9}$$

and

$$a_0 = \frac{2}{N_0} \sum_{k=0}^{N_0-1} x(k), \tag{10}$$

$$a_1 = \frac{2}{N_0} \sum_{k=0}^{N_0-1} x(k) \ \cos\left(2\pi\frac{k}{N_0}\right), \tag{11}$$

$$b_1 = \frac{2}{N_0} \sum_{k=0}^{N_0-1} x(k) \ \sin\left(2\pi\frac{k}{N_0}\right). \tag{12}$$

For the above DFS expression formula, $N_0$ must be an integer, and only synchronous sampling can satisfy the condition, which can be expressed as follows [15]:

$$f_s = N_0 \times f_0 \ or \ T_0 = N_0 \times t_s. \tag{13}$$

In practice, sampling is usually asynchronous. Thus, the reconstruction of the signal using the DFS algorithm can result in deviations, which are largely influenced by the asynchronous factor $\alpha$ ($|\alpha| < 0.5 \ and \ \alpha \neq 0$). The correlation between sampling rate $f_s$ and signal frequency $f_0$ for asynchronous sampling is as follows [13,14]:

$$f_s = (N_0 + \alpha)f_0, \tag{14}$$

$$N_{DFS} = N_0 = round\left(\frac{f_s}{f_0}\right). \tag{15}$$

To attenuate the effect of deviations caused by the asynchronous factor $\alpha$ in asynchronous sampling, a recursive self-correction (RSC) algorithm based on the DFS algorithm is indispensible [7].

### 2.5. Recursive Self-Correction Algorithm

The self-correction algorithm (see Figure 3) seeks to reduce the deviations in the calculated coefficients a0, C1, and Φ1 of the Fourier series by reconstructing the signal and then calculating the Fourier coefficients from the reconstruction signal. Then, the deviations of the coefficients can be derived as follows [13,14]:

$$\Delta Y = Y_r - Y^* \ with \ Y = a_0, \ c_1, \Phi_1, \tag{16}$$

where $Y$ represents the original coefficients, and $Yr$ stands for the reconstruction signal's calculated coefficients. This process is called self-calibration and is developed primarily for data processing of asynchronously sampled signals. The deviations can be subtracted from the initial coefficients, so that the result becomes more accurate after each self-correction iteration [13,14]:

$$Y = Y^* - \Delta Y = 2Y^* - Y_r \quad with \ Y = a_0, \ c_1, \Phi_1. \tag{17}$$

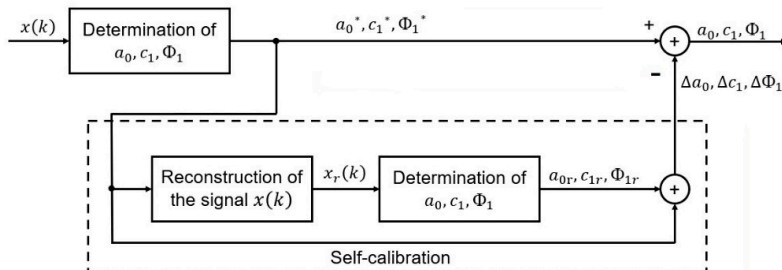

**Figure 3.** Principle of the self-correction algorithm of the coefficients $a_0$, $c_1$, and $\Phi_1$ [13].

For obtaining the more accurate coefficients $Y$, the self-correction should be performed several times sequentially. This process is called the iterative self-correction (RSC) algorithm. With iterative self-correction performed twice in succession (see Figure 4), the coefficients $a_0$, $c_1$, and $\Phi_1$ are determined as follows [13,14]:

$$Y = Y_2 = 3Y^* - (Y_{r1} + Y_{r2}) \quad with \ Y = a_0, \ c_1, \Phi_1, \tag{18}$$

where $Y_{r1}$ and $Y_{r2}$ represent the coefficients using the reconstructed dataset in the first and second self-corrections.

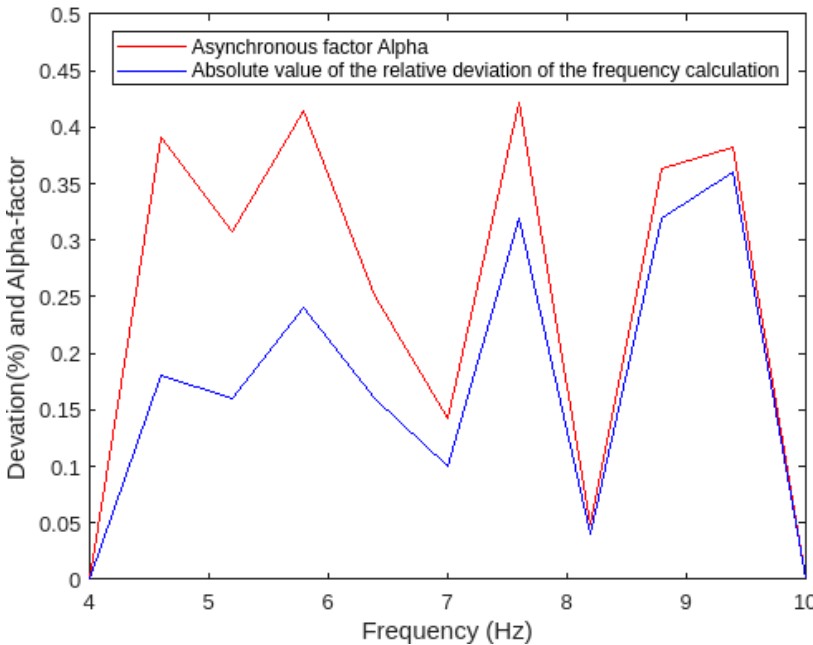

**Figure 4.** Principle of the iterative self-correction algorithm of the coefficients $a_0$, $C_1$, and $\Phi_1$ performed twice in a sequence [13].

In general, the coefficients in the algorithm with J iterative self-corrections can be represented by the following expression [13,14]:

$$Y = Y_J = (J+1)Y^* - \sum_{j=1}^{J} Y_{rj} \quad with \ Y = a_0, \ c_1, \Phi_1, \tag{19}$$

where $Y_{rj}$ ($j = 1, 2, \ldots, J$) are the coefficients using the reconstructed dataset in the j-th self-calibration.

### 3. Verification of the New Method Based on Simulation in MATLAB

The SWPM method, which can be implemented in an MCU, can precisely calculate the initial frequency of the low-speed signal $f_0$. Thus, the DFS and RSC algorithms were simulated on MATLAB to verify the feasibility of using the RSC-DFS method and frequency measuring method based on phase difference.

Without considering noise or assuming a very high SNR, the sampling signal can be described by Equation (7) (see Section 2.4). Through this equation, discrete sine signals can be simulated and reconstructed by the DFS algorithm in MATLAB. After the signal reconstruction, significant signal parameters are calculated, such as the phase $\Phi_1$. The phase difference $\Delta$ can be determined by two reconstructions at time intervals of $t_s$. According to Equation (6) (see Section 2.3), the signal frequency can be calculated. To verify the algorithm, the sampling rate $f_s$ is set to 1000 Hz, and the sampling period $t_s$ is set to 1 ms, while the signal frequency ranges from 4 to 10 Hz. The frequency calculation results of the DFS algorithm and the asynchronous factors $\alpha$ are visualized in Figure 5.

**Figure 5.** Measuring deviation of the DFS algorithm and the asynchronous factor $\alpha$.

The correlation coefficient $r_{dev,\alpha}$ is calculated for the deviations *dev* and the asynchronous factors $\alpha$ presented in the above graph [16].

$$r_{dev,\alpha} = \frac{cov(dev,\ \alpha)}{\sqrt{var(dev) \times var(\alpha)}}. \tag{20}$$

A closer absolute value of the correlation coefficient to 1 indicates a stronger degree of linear correlation between the two variables [16]. The MATLAB simulation yields a $r_{dev,\alpha}$ equal to 0.9120, which is close to 1. This indicates that the asynchronous factor has a highly linear correlation with the frequency calculation deviation of the DFS algorithm. Since asynchronous sampling cannot be avoided, the RSC algorithm is necessary to mitigate the asynchronous factor's impact. The results from the frequency simulation measurements of the RSC-DFS algorithm are shown in Figures 6 and 7. The results show that the RSC algorithm can effectively suppress the asynchronous factor's impact and greatly improve the measuring accuracy. Additionally, multiple RSCs can theoretically further improve the measuring accuracy. The relative error almost disappears after two iterations of the RSC, resulting in a theoretical deviation of about 0. However, the number of RSC iterations increases the calculation time. To avoid the frequency calculation time exceeding the signal sampling period, the number of RSC iterations should be minimized. Usually, two iterations are sufficient.

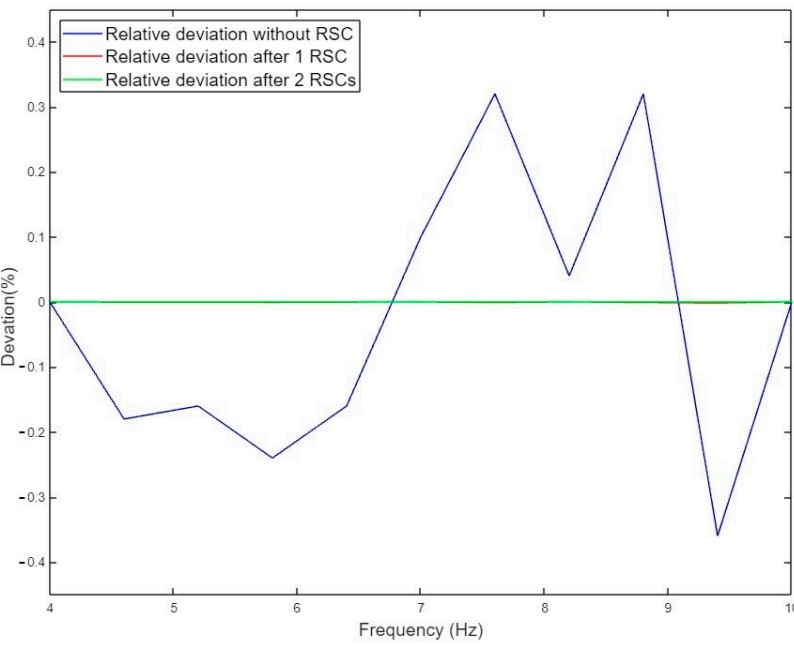

**Figure 6.** Improvement of DFS frequency measuring method using the RSC algorithm.

The above simulation results are based on noise-free signals or signals with high SNR. Yet there are always noise disturbances. Hence, the simulation results of signals with different SNRs were evaluated. Similarly, each signal group with the same SNR is divided into 10 equal parts with frequency ranging from 4 to 10 Hz. Thus, each group had 11 test points, which were then simulated under different usages of the RSC algorithm. The root-mean-square error (RMSE) analysis was performed for each group of measurements; considering the randomness of the noise, each group was also simulated 10 times. Afterward, the RMSE values of the 10 different measurement groups were averaged. Furthermore, the asynchronous factor's impact on the frequency measurement's deviation without self-correction of the DFS algorithm signals was investigated under different SNRs. The simulation results are shown in Table 2.

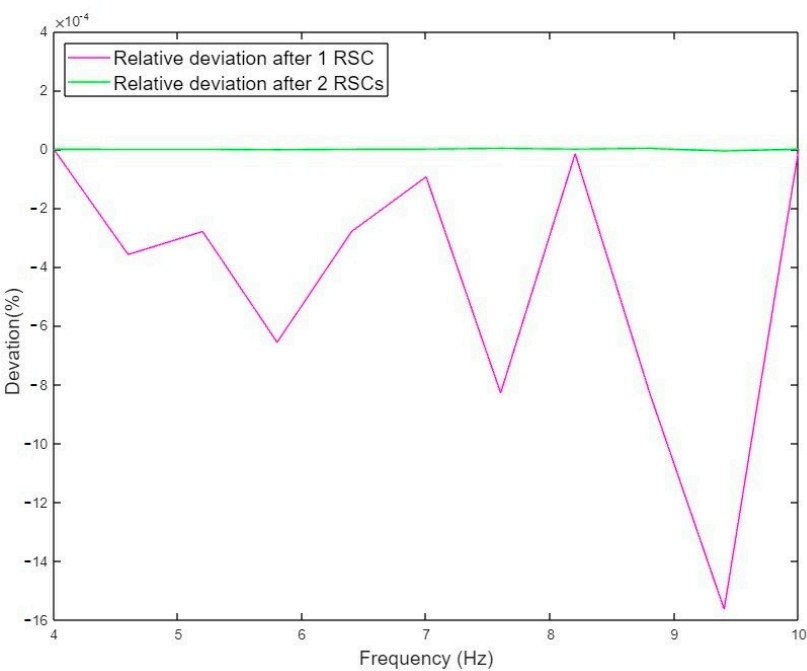

**Figure 7.** Improvement of measuring results using the iterative RSC.

**Table 2.** Simulation results of signals with different SNR in MATLAB.

| SNR (dB) | $r_{dev,\,\alpha}$ [1] | RMSE [2] of Dev without RSC (%) | RMSE [3] of Dev after 1 RSC (%) | RMSE [4] of Dev after 2 RSCs (%) |
|---|---|---|---|---|
| 20 | −0.1278 | 2.166 | 2.138 | 2.138 |
| 30 | 0.1821 | 0.7635 | 0.7263 | 0.7262 |
| 40 | 0.3080 | 0.2620 | 0.2062 | 0.2062 |
| 50 | 0.7895 | 0.2146 | $6.675 \times 10^{-2}$ | $6.671 \times 10^{-2}$ |
| 60 | 0.8844 | 0.2099 | $2.239 \times 10^{-2}$ | $2.245 \times 10^{-2}$ |
| 70 | 0.9062 | 0.2121 | $6.829 \times 10^{-3}$ | $6.798 \times 10^{-3}$ |
| 80 | 0.9115 | 0.2105 | $2.206 \times 10^{-3}$ | $2.180 \times 10^{-3}$ |
| 90 | 0.9117 | 0.2105 | $9.262 \times 10^{-4}$ | $6.288 \times 10^{-4}$ |
| 100 | 0.9119 | 0.2105 | $6.835 \times 10^{-4}$ | $2.230 \times 10^{-4}$ |
| 110 | 0.9120 | 0.2104 | $6.506 \times 10^{-4}$ | $7.260 \times 10^{-5}$ |

[1][2][3][4] Average of measured values after 10 simulations for each group with the same SNR.

According to the correlation coefficients in the table, the $r_{dev,\,\alpha}$ value is greater than 0.2 when the SNR is higher than 40 dB. This means that the asynchronous factors $\alpha$ and the measuring deviations *dev* are correlated. A higher SNR resulted in greater correlation. In addition, the RMSE values of the relative deviations indicate that the implementation of the RSC algorithm only improved the measuring accuracy significantly when the SNR was higher than 50 dB. According to the simulation results, the SNR should be greater than 30 dB to achieve a measuring deviation of less than 1%. If the measuring deviation is less than 0.1%, the corresponding SNR should be greater than 50 dB.

## 4. Practical Results of the New Method

### 4.1. Measuring Result Based on Signal Generator

To verify the performance of the RSC-DFS algorithm in practical applications, the microcontroller-based signal processing unit CYSPU-98A in Chenyang Technologies GmbH was developed. It improves the real-time performance of low-speed measurements through cooperative digital processing of square wave and sine wave signals. This device mainly consists of a microcontroller signal processing module, an RS485 communication module,

and an LCD display module. Three input modes are classified according to the input signal: sine input only, square wave input only, and square wave plus sine input [17]. This paper only uses sine and square wave signals to study low-speed measuring methods. Therefore, only functions relevant to these signals are used. The CYSPU-98A unit applies this novel low-speed measurement method, using two iterations of the RSC algorithm, to improve the measuring accuracy.

In this article, square and sine wave signals generated by the signal generator JDS-2900 were used as input signals of the signal processing units. The signal generator has a frequency accuracy of $\pm 20$ ppm, and the power ratio between the sine signal and the carrier signal can be up to 40 dB. The block diagram of the signal generator-based test system is visualized in Figure 8.

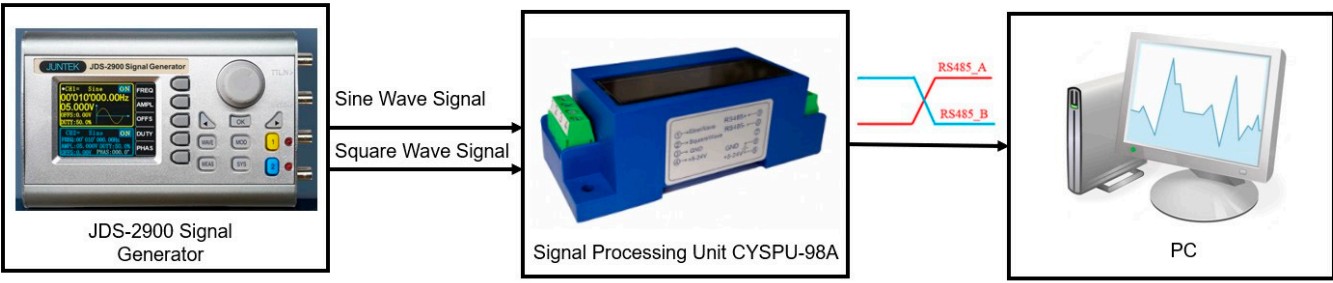

**Figure 8.** Block diagram of the signal generator-based frequency test system.

The measuring signals in the frequency range of 0.4 Hz to 20 Hz are generated by the signal generator and used as input of the signal processing unit for frequency calculation. One hundred values are measured for each group of frequency values. Afterward, the triple coefficient of variation $\frac{3\sigma}{\mu}$ and the absolute mean relative devation are calculated to evaluate the measuring accuracy. The measuring results are shown in Figure 9.

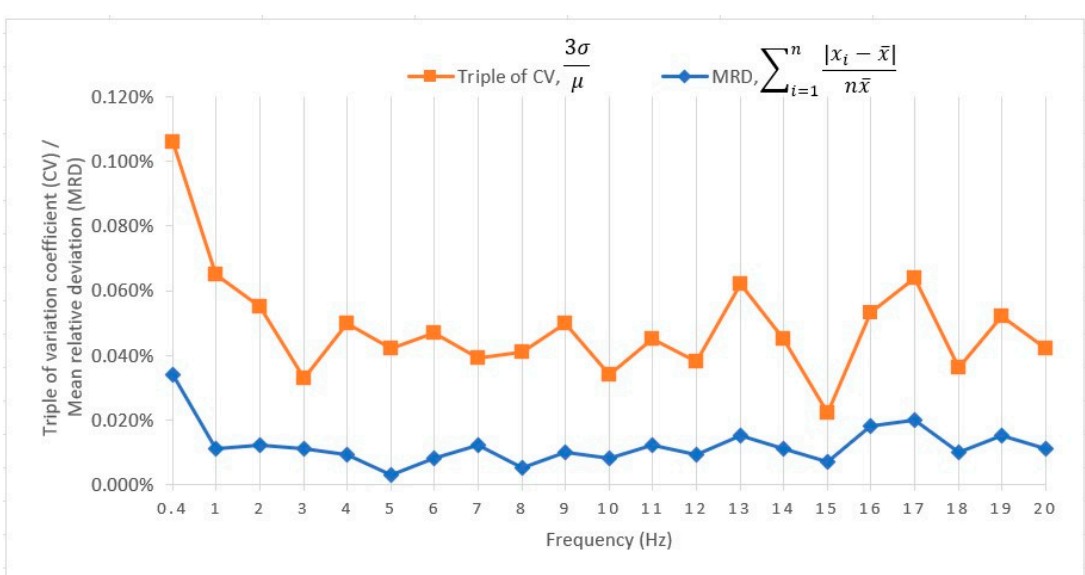

**Figure 9.** Frequency measuring results of signals generated by signal generator.

The above figure shows that the signal processing unit with RSC-DFS algorithm has a measuring error of about 0.1% when the signal input is provided by the signal generator. According to Section 3, the theoretical measuring error depends on the SNR of the sine signal. Considering that the SNR of the generator's output signal is greater than 40 dB, the measuring error of 0.1% is as expected.

### 4.2. Measuring Result Based on Hall Effect Gear Tooth Sensor Speed Measurement System

A Hall effect gear tooth speed sensor consists of a permanent magnet, a linear Hall element, and a target iron gear (see Figure 10). The Hall element is a magnetic field sensor that converts a magnetic flux density linearly into an output voltage. When the target wheel rotates periodically at a suitable sensing distance, the magnetic flux through the Hall element is periodically changed. The Hall element detects the magnetic flux change and outputs a corresponding periodic voltage, which experimentally proves to be a sinusoidal signal [18].

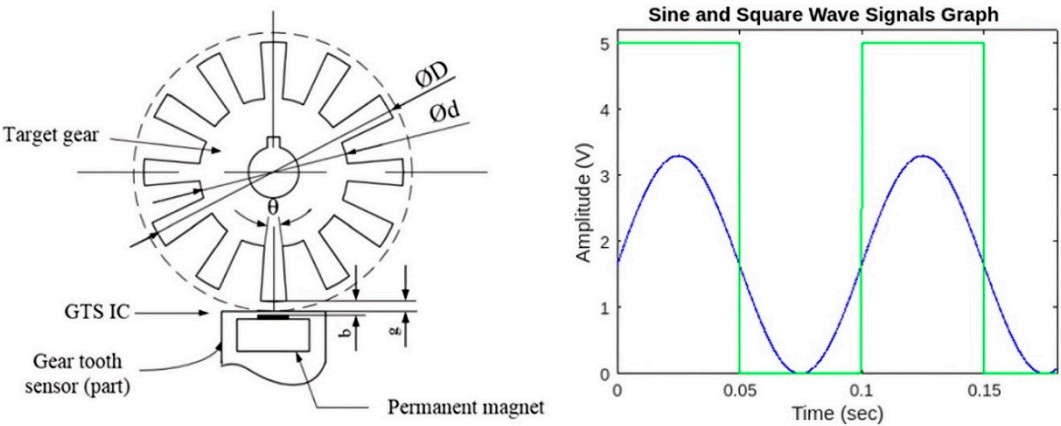

**Figure 10.** Hall effect gear tooth sensor CYGTS102DC and its output signals.

The sinusoidal output signal of the Hall element is amplified and then converted to a square wave signal by a comparator. The sine and square waves have the same frequency (see Figure 10), which is a requirement of the novel low-speed measuring method. This type of Hall effect sensor was developed by Chenyang Technologies GmbH and is called CYGTS102DC. This specially designed gear tooth sensor with a biasing magnet and internal denoising filter is sealed in resin for physical protection and cost-effective installation. Two signals (one sinusoid wave and one square wave) are outputted directly through the output terminal of the operational amplifier. This sensor operates with a good signal-to-noise ratio and excellent low-speed performance [19].

In a Hall effect sensor speed measuring system, a gear with $N_r$ teeth will generate $N_r$ signal cycles for each revolution. Therefore, the relationship between signal frequency and rotational speed can be derived as follows [15]:

$$\omega = \frac{60f}{N_r} (\text{rpm}). \tag{21}$$

In the experiment, the Hall effect sensor CYGTS102DC and signal processing unit CYSPU-98A were combined to perform low-speed measurement tests. In the speed measuring system, a high-resolution servo motor SGM7J-02AFC65 was used to drive the rotation of the gear shaft. The speed was set by computer software as a reference speed value. The Hall effect sensor detected the gear movement and outputted both sinusoidal and square signals. Then, the signal processing unit processed the two signals to determine the rotational speed of the servo motor. The block diagram of the Hall Effect gear tooth sensor-based test system is shown in Figure 11.

In the speed measuring system, a gear with 12 teeth was used as the target gear. In this case, the frequency range from 0.4 to 20 Hz corresponds to a rotational speed range of 2–100 rpm (see Equation (21)). Speed measurements were performed in the speed range of 2 to 100 rpm by setting the speed in the software that controls the servo motor. Similarly, $\frac{3\sigma}{u}$ and the absolute mean relative deviations of 100 test data in each group were calculated. Figure 12 shows the measuring results in the low-speed measurement.

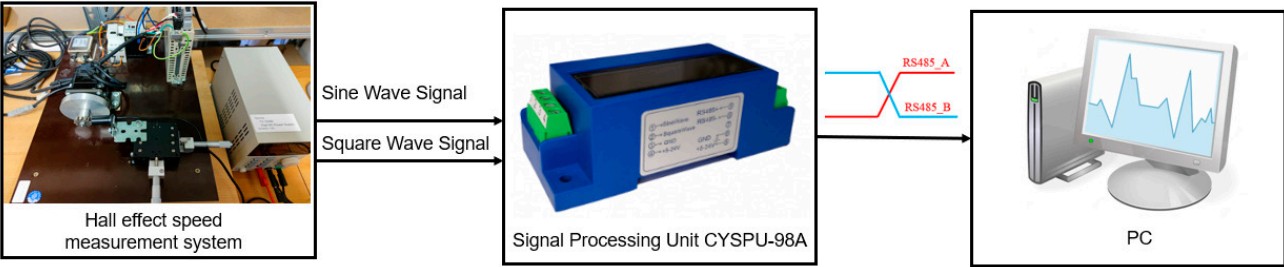

**Figure 11.** Block diagram of a Hall effect gear tooth sensor-based frequency test system.

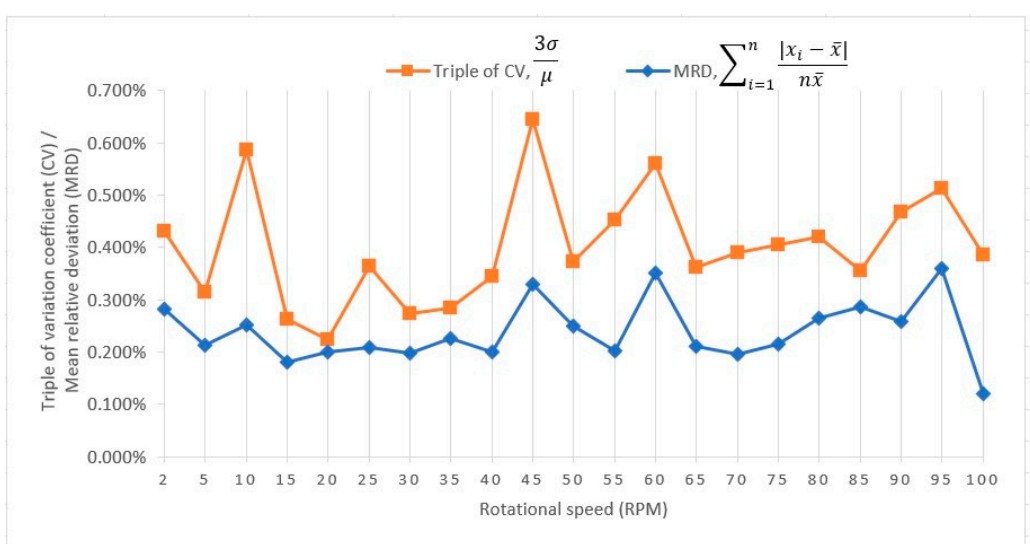

**Figure 12.** Frequency measuring results based on Hall effect gear tooth sensor.

Figure 12 shows that the measuring error using a Hall effect gear tooth sensor was less than 1%. Compared to the signal generator, the maximum SNR of the analog sine signal of the Hall effect gear tooth sensor was only about 30 dB; therefore, the measurement error based on the Hall effect gear tooth sensor speed measurement system became larger due to the signal noise.

Furthermore, this new low-speed measuring method can be applied to any speed sensors that can provide sinusoidal and square wave signals of the same frequency. For example, the method can be used with the optical reflective gear tooth sensor CYGTS102OR and the optical transmission circular grating sensor CYRSS102OG of Chenyang Technologies GmbH & Co. KG.

### 4.3. Measuring Results on Motor Variable Speed Motion

In the Hall effect gear tooth sensor speed measuring system, the servo motor is used to drive the target gear in a periodic motion where the motor's speed can be controlled by software. The low-speed measuring system in Section 4.2 was subsequently used for variable speed measurements. The motor was controlled by software to perform variable speed movements with a speed range from 0 to 100 RPM as shown in the speed profile in Figure 13. The motor speed was first constantly accelerated to 100 RPM, and then kept constant for a certain time before decelerating it to standstill. The process was repeated periodically.

The signal processing unit CYSPU-98A used the RSC-DFS algorithm method for the low-speed measurement mentioned in this paper. The algorithm speeds up the low-speed frequency calculation with a high output rate of calculated values, which is equal to the sampling rate of the sinusoidal signal $f_s$. This unit's data output is approximately 256 values per second (vps) for frequencies less than 1 Hz and 512 vps

for frequency from 1 Hz to 25 Hz [17]. The speed measuring curve of the measuring system is shown in Figure 14.

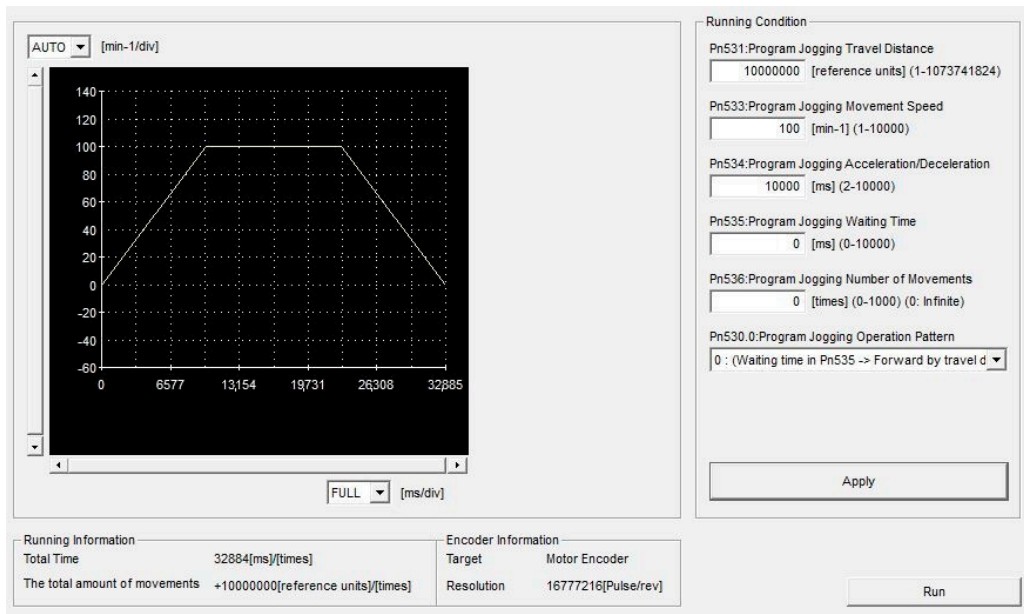

**Figure 13.** Speed curve of the servo motor SGM7J-02AFC65.

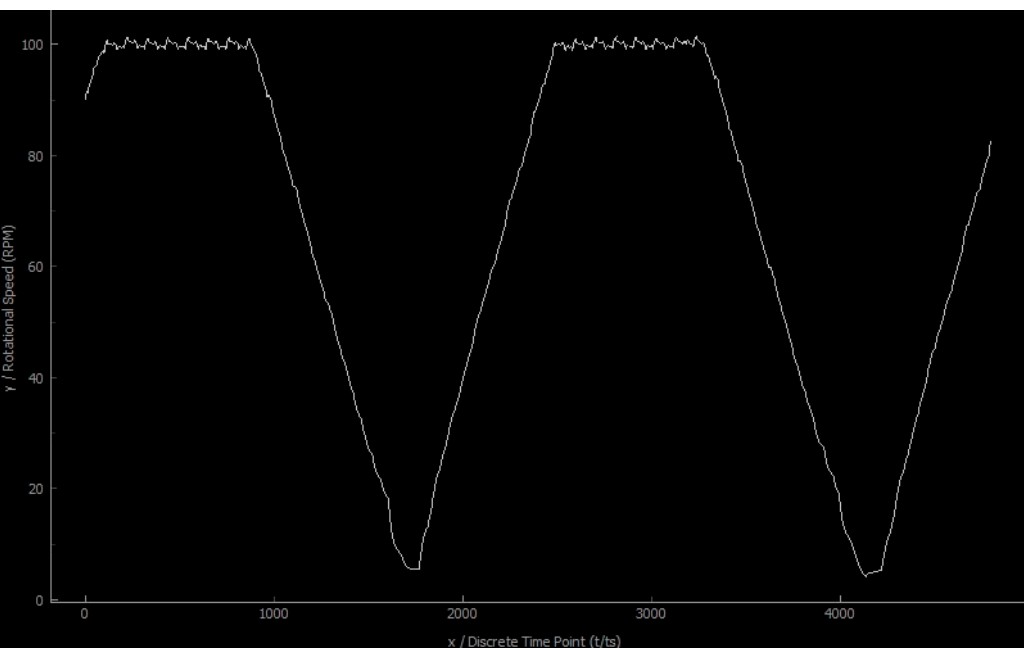

**Figure 14.** Speed curve of the measuring results based on the Hall effect gear tooth sensor CYGTS102DC and the signal processing unit CYSPU-98A.

In the case of traditional SWPM method, the output rate's upper limit is the frequency of the measured signal *f*. For example, in a frequency range from 0 to 25 Hz, the maximal output rate for the SWMP method is 25 vps. The minimum value of the output rate ratio $\mu$ for the two low-speed measuring methods can be derived by division.

$$\mu_{min} = \frac{f_s}{f_{max}} = \begin{cases} \frac{512\text{vps}}{25\text{vps}} = 20, & \text{for } \mathbf{1Hz} \leq f \leq \mathbf{25Hz} \\ \frac{256\text{vps}}{1\text{vps}} = 256, & \text{for } \mathbf{0} < f < \mathbf{1Hz} \end{cases}. \tag{22}$$

The novel measuring method's output rate for frequencies from 1 Hz to 25 Hz is more than 20 times that of the traditional measuring method. For frequencies less than 1 Hz, it is more than 256 times, which significantly improves the response time and real-time performance of low-speed measurements. The presented low-speed measuring method allows the monitoring of low and variable speed motions in a shorter response time.

## 5. Potential Application Examples

### 5.1. Direct-Drive Offshore Wind Turbine

In order to alleviate the global energy crisis, the development and exploitation of renewable energy sources such as wind energy have become a research focus.

Wind turbines such as land-based gearbox turbine and direct-drive offshore wind turbine are the fundamental equipment to obtain wind energy. The critical difference between the two types of wind turbines lies in the principle of power generation. Direct drive turbines have no gearboxes, which simplifies the nacelle system and increases efficiency, as well as reliability. They work by connecting a slowly moving rotor directly to a generator to produce electricity. In contrast to onshore wind turbines, where the high-speed shaft's speed behind the gearbox can be measured, offshore wind turbines can only measure the speed of the low-speed shaft. The direct-drive wind turbine without gearbox has many advantages such as high efficiency at low wind speed, low noise, high lifetime, reduced unit size, and lower operation and maintenance cost. However, as there is no gearbox in the generator set, the speed range of the generator and shaft, as well as the accuracy of the speed measurement using encoders, is low. For this issue, the low-speed measuring method mentioned in this paper can be an alternative.

Figure 15 presents the working principle of a direct-drive wind turbine. Here, the rotor speed is an important input variable for the main control system and the converter control system [20]. The speed measurement of the low-speed shaft must meet not only high accuracy, but also good real-time performance. Thus, the damage to the equipment caused by turbine speed disorder can be avoided. In this case, a speed sensor such as the Hall speed sensor CYGTS102DC, which can output both square wave and sine signals, needs to be mounted adjacent to the low-speed shaft. When the wind wheel rotates the low-speed shaft, the speed sensor outputs a speed signal, which is continuously measured by a signal processing unit such as the CYSPU-98A. When the wind turbine speed changes significantly, the corresponding signal processing unit responds rapidly by outputting the actual speed value, thus allowing the control systems to react quickly to avoid overload.

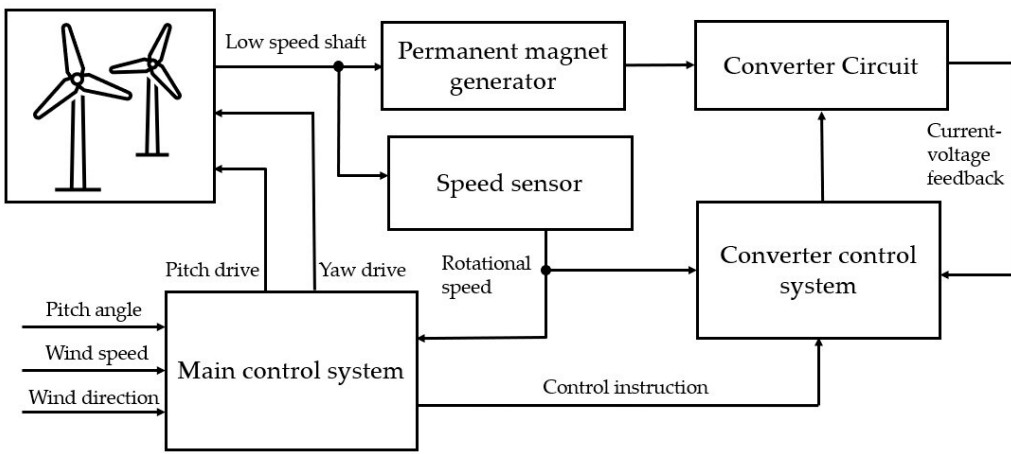

**Figure 15.** Schematic diagram of direct drive permanent magnet wind turbine.

### 5.2. Precision Low-Speed Motor Controller

Precision motors are usually quite expensive and inefficient, when they operate at slow speeds. Innovators at the NASA Johnson Space Center have developed a method to control precise motion of a brushless DC (BLDC) motor using relatively inexpensive

components. Current motors are only able to operate at approximately 15 RPM with a risk of excessive jitters. This technology reduces the responsive RPMs by several orders of magnitude to approximately 0.025 RPM. Its ability to operate at these ranges with high precision provides an opportunity to integrate this technology into many applications and industries [21]. It uses BLDC motors to achieve low-speed precision control. A BLDC motor control system is shown in Figure 16.

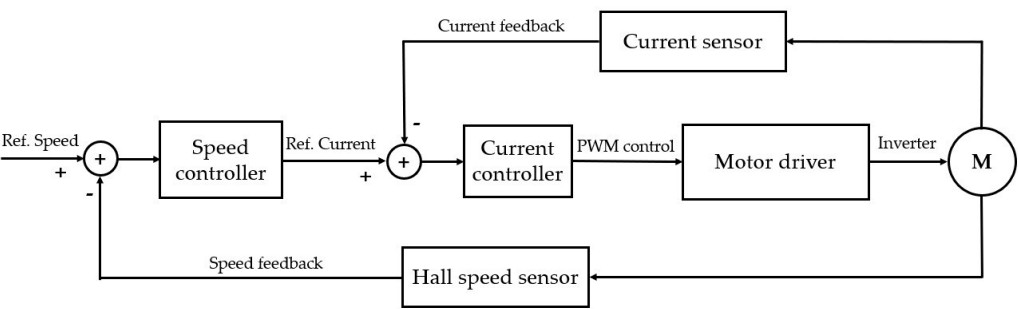

**Figure 16.** Block diagram of the BLDC motor control system.

According to Figure 16, the speed signal of the BLDC motor can be detected by Hall effect speed sensors, while the signal processing of the speed signal is essential for the next speed control step [22]. In these parts of the speed measuring process, the presented low-speed measuring method can be applied. For example, the Hall effect speed sensor CYGTS102DC and the signal processing unit CYSPU-98A can be combined to measure the rotor speed of a BLDC motor during low-speed motion. Then, the speed and current information is fed back for the PWM control, which in turn controls the inverter bridge to change the winding's power-up state. When the motor works at low speed, accurate and high real-time speed measurement improves the accuracy of precision control.

## 6. Conclusions and Suggested Future Work

In this paper, a novel low-speed measuring method based on the sine and square signals of a signal generator or a Hall Effect gear tooth speed sensor was presented. This method focuses on accurate and low-cost measurement in low-speed ranges.

Considering the asynchronous sampling of the ADC module, the novel method adopts the RSC-DFS algorithm to correct the effect of asynchronous factors. In MAT-LAB simulations, multiple RSCs can effectively reduce the measurement error. Even a theoretical error equal to zero has been achieved. However, all frequency calculations need to be performed within one sampling period because of the sequential execution of MCUs. Thus, two RSC iterations are usually sufficient with the tradeoff between the calculation's time cost and the measuring accuracy.

The experimental results verified the effectiveness of the proposed method. Compared to the conventional SWPM method at low-speed measurements, this novel method guarantees a high measuring accuracy and still provides a faster measuring rate to improve the system's real-time performance. As a practical implementation, the signal processing unit CYSPU-98A was successfully developed by Chenyang Technologies GmbH & Co. KG.

In combination with various gear speed sensors, this measuring method is able to provide promising results in the field of low-speed measurement applications such as direct-drive offshore wind turbines, as well as precision low-speed motor controllers. It achieves fast and accurate and low-cost measurements; thus, it has great applicative potential in the industrial low-speed measuring field. Future research will focus on the usage of field-programmable gate array (FPGAs) as an alternative to MCUs. The FPGAs can execute programs in parallel; thus, more RSCs can be executed to improve

measuring accuracy and provide a larger measurement range because of the sampling time's independence.

**Author Contributions:** Conceptualization, J.L.; methodology, Q.S. and J.L.; software, Q.S.; validation, Q.S.; formal analysis, Q.S. and J.L.; investigation, Q.S.; resources, Q.S. and J.L.; data curation, Q.S.; writing—original draft preparation, Q.S.; writing—review and editing, Y.L. and J.L.; visualization, Y.L. and J.L.; supervision, J.L.; project administration, J.L. All authors have read and agreed to the published version of the manuscript.

**Funding:** This research received no external funding.

**Data Availability Statement:** Restrictions apply to the availability of these data. Data were obtained from ChenYang Technologies GmbH & Co.KG and are available from Y.L. with the permission of ChenYang Technologies GmbH & Co.KG.

**Conflicts of Interest:** Q.S. is a working student of ChenYang Technologies GmbH & Co.KG; J.L. is the technical manager of ChenYang Technologies GmbH & Co. KG; Y.L. is an employee of ChenYang Technologies GmbH & Co. KG.

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
