# Peer review of "Novel Low-Speed Measuring Method Based on Sine and Square Wave Signals"

_2673-8244, doi:10.3390/metrology3010005_

Round 1
Reviewer 1 Report
The paper titled "Novel low-speed measuring method based on sine and square wave signals" describes a research contribution made by the Authors to the metrology field mainly focused to the low-speed measurement methods.
The paper merits are summarized: (i) clear aims and structure, (ii) graphics and data, and (iii) method presentation.
However, prior to accept this manuscript to publication, several changes that the Authors have to:
1. On Section 5 - Potential application examples, for each example, a concrete use case should be presented.
Best Regards,
The Reviewer
Reviewer 2 Report
The authors present a novel method to measure low rotational speeds based on analog sine and square waves of Hall Effect speed sensors coupled with correlative digital signal processing algorithms.
The paper is well written and logically organized. The proposed method is clearly presented, and the reported results validated the approach. For these reasons I recommend the paper for publication.
Author Response
Thank you very much for your review
Reviewer 3 Report
Review of the manuscript metrology-2115168
Novel low-speed measuring method based on sine and square wave signals
Journal
Metrology (ISSN 2673-8244)
Manuscript ID
metrology-2115168
he paper presents a novel approach to measuring low speeds using analog sine and square waves of Hall Effect speed sensors coupled with correlative digital signal processing algorithms packaged on a signal processing unit. My comments on the manuscript are as follows:
1- The introduction part of the manuscript is weak, particularly the literature review section.
2- You also have to compare it with the reported literature.
3- Make a table to indicate all the results in a comparative manner. To evaluate the comparison of the mentioned methods according to their simulation speed signals and measurement of a time interval between successive pulses. In addition to the counting of pulses during the prescribed time. and measurement of the time duration for the variable number of pulses for the proposed system. The impacts of the quantization error by using a large number of periods in the speed calculations. The comparisons and validations of the Recursive self-correction (RCS) principle applied to the accuracy improvement of the S-DFT can be discussed. The advantages of the proposed methods for square wave period measuring method (SWPM) Recursive self-correction (RSC) algorithm is used to perform the low-frequency measurement using the discrete sinusoid wave proposed compared with the existing methods which significantly increases the theoretical and practical value of the methodology on this the existing approaches and proposed statistical hypotheses.
4- In the experiment, the Hall Effect sensor CYGTS102DC and signal processing unit CYSPU-98A are combined to perform low-speed measurement tests. In the speed measuring system, a high-resolution servo motor SGM7J-02AFC65 is used to drive the rotation of the gear shaft and the speed can be set by computer software as a reference speed 346 value. The Hall Effect sensor detects the movement of the gear and outputs both sinusoidal and square signals. Explain in more detail the importance, description, and accuracy of CYGTS102DC and signal processing unit CYSPU-98A. The discussion on the obtained results needs to be increased for a better understanding of the reader about the significance of utilization of the proposed method and limitations of models because of complex computational.
There is no specific comparison of the method with previously known research results. Some simulation or experiment results of comparison should be added. The comparison for high accuracy and
bandwidths, increased voltage margin, great dynamic response, robustness, etc. as will be verified experimentally in the proposed design.
5- What is the reason the measuring principle is limited due to the long counting time T of at least one signal period and does not meet the requirements for high real-time performance?
6- Despite the high measuring accuracy, the measuring principle is limited due to the long counting time T of at least one signal period and does not meet the requirements for high real-time performance.
the applications of the discrete Fourier series (DFS) Algorithm. The proposed method should be compared with some well-known methods in this area.
7- Please provide more details concerning and reference for the equations in the manuscript.
8- What Would be the implication for the practical implementation of the RSC-DFS algorithm on the sine and square wave signals for the proposed scheme in the other application and illustrate more about the application of direct-drive offshore wind turbine? The results need more elaboration and may also be presented in tabular form for better clarity. The obtained results are to be explained properly and in detail with the optimized parameters.
9- Modify the abstract and conclusion. The discussion is very too long. Please reduce the discussion. The conclusion section must describe in detail the major finding of the paper.
10- Since asynchronous sampling cannot be avoided, the RSC algorithm is necessary to mitigate the asynchronous factor’s impact. Investigate the steps and process in the RSC to mitigate the asynchronous factor’s impact RSC algorithm can effectively suppress the impact of the asynchronous factor and improve the measuring accuracy and how do resulting in a theoretical deviation of about 0.
11- A thorough revision of the English language usage is required for improving the technicality of the paper.
12- The references list needs to update, there are many important references regarding this field that are not addressed in the literature review of this work. There are old references. There are many references from 1996, 2000, 2001, 2003, 2007 2008, and 2010. Please provide references for this topic in 2021, and 2022. The topic is old and published in previous research. Relevant parameters and dimensions with appropriate references are to be included
13- Please a new section for related work and consider the following works:
*My recommendation
Major revision
Reviewer 4 Report
In this study, hall effect sensor based low speed measuring method were presented. Topic is interesting but some minor revisions should be done. The abstract is sufficiently informative. Comments are given below:
1. The abstract is sufficiently informative. Your achievements should be stressed with the novelty (RSC) using your results. State of the art of the study should be clearly defined in the paper for readers.
2. There is no conclusion in your paper. Please give a separate conclusion section in the paper.
3. Your future aspects should be given in details in Conclusion. Maybe some potential application areas should be given in this section.
4. There are too many self-citation is given please remove, if not necessary.
Round 2
Reviewer 3 Report
i- The authors have responded to individual queries asked by the reviewer. But the curves are not clear enough and need improvement in accuracy and lines in Figures (6), (12), and (14). The graph's resolution and fonts could be better in the Figures aforementioned.
ii- The paper abstract is long. Could you please shorten the abstract a bit?
Would it be possible for the authors to shorten it and highlight which phenomenon is the most important to get the system's performance improved? Also, in Section "Conclusions " is so long 28 lines. Reduce them and change the title Conclusions and suggested future work.
iii- The figure and resolutions aren't clear. Figures (6) and (12) should be individual for the clarification of the achieved results. Figs. (9) and (10) must be enlarged and clear because they are in the context of the experiment for this proposed work.
iv- The references should be of good quality and up to date. There are old references such as 1996, 2000, 2006, and 2010. Please provide a new reference for the years (2022) and (2023). References are not good, not recent, and are all old works
v- Provide abbreviation microcontroller (MSC).
vi- The conclusions part must be improved with the aid of drawing comparisons with similar works across the literature.
vii- A plan of practical implementation needs to be included.
viii- Write a description of Figure 1 directly below figure 1--it's written at the beginning of the next page.
iv- I recommend accepting and publishing the research paper after these minor comments mentioned above
* My Recommendation
Accept after minor revisions
